# Regulators of the Asexual Life Cycle of *Aspergillus nidulans*

**DOI:** 10.3390/cells12111544

**Published:** 2023-06-04

**Authors:** Ye-Eun Son, Jae-Hyuk Yu, Hee-Soo Park

**Affiliations:** 1Major in Food Biomaterials, School of Food Science and Biotechnology, Kyungpook National University, Daegu 41566, Republic of Korea; thsdpdms0407@naver.com; 2Department of Bacteriology, Food Research Institute, University of Wisconsin-Madison, Madison, WI 53706, USA; jyu1@wisc.edu; 3Department of Integrative Biology, Kyungpook National University, Daegu 41566, Republic of Korea

**Keywords:** *Aspergillus nidulans*, asexual development, conidial dormancy, conidial germination

## Abstract

The genus *Aspergillus*, one of the most abundant airborne fungi, is classified into hundreds of species that affect humans, animals, and plants. Among these, *Aspergillus nidulans*, as a key model organism, has been extensively studied to understand the mechanisms governing growth and development, physiology, and gene regulation in fungi. *A. nidulans* primarily reproduces by forming millions of asexual spores known as conidia. The asexual life cycle of *A. nidulans* can be simply divided into growth and asexual development (conidiation). After a certain period of vegetative growth, some vegetative cells (hyphae) develop into specialized asexual structures called conidiophores. Each *A. nidulans* conidiophore is composed of a foot cell, stalk, vesicle, metulae, phialides, and 12,000 conidia. This vegetative-to-developmental transition requires the activity of various regulators including FLB proteins, BrlA, and AbaA. Asymmetric repetitive mitotic cell division of phialides results in the formation of immature conidia. Subsequent conidial maturation requires multiple regulators such as WetA, VosA, and VelB. Matured conidia maintain cellular integrity and long-term viability against various stresses and desiccation. Under appropriate conditions, the resting conidia germinate and form new colonies, and this process is governed by a myriad of regulators, such as CreA and SocA. To date, a plethora of regulators for each asexual developmental stage have been identified and investigated. This review summarizes our current understanding of the regulators of conidial formation, maturation, dormancy, and germination in *A. nidulans*.

## 1. Introduction

Filamentous fungi are eukaryotic organisms that are ubiquitous in our surrounding environments, and they have large and small effects on human life [1,2,3]. Several fungi, such as *Fusarium*, *Aspergillus*, and *Penicillium*, produce mycotoxins that can cause plant diseases or contaminate stored foods, leading to economic losses [4]. Some fungi can directly exert adverse clinical effects on humans [1]. Conversely, other filamentous fungi have been used as industrial cell factories for producing various proteins because of their efficient secretion and sustainable production systems [5,6]. Therefore, to suppress side effects or maximize positive effects, it is important to understand the biological characteristics of filamentous fungi.

*Aspergillus* is one of the filamentous fungi that comprise the maximum proportion of airborne organisms. Among the numerous species, *Aspergillus flavus* is a saprophytic fungus that contaminates preharvest and postharvest crops and produces potent hepatocarcinogenic secondary metabolite aflatoxins; it is known to be the second leading cause of invasive aspergillosis [7,8]. *Aspergillus fumigatus* is an opportunistic pathogenic fungus that causes life-threatening disease (aspergillosis) in immunocompromised individuals [9,10]. In contrast, *Aspergillus niger* and *Aspergillus oryzae* are biochemical cell factories in the fermentation and enzyme industry, which efficiently produce several enzymes and useful secondary metabolites [11,12]. These *Aspergillus* species have some limitations in controlling and handling for research; therefore, many scientists have explored the specie *Aspergillus nidulans* for decades as a referential model organism to uncover fungal growth, asexual/sexual development, spore properties, germination, and secondary metabolites [13,14,15].

The asexual spore (conidium) is the primary reproductive structure with long-term viability and contains various secondary metabolites including mycotoxins [14,16,17]. Conidia, which float through the air, settle on crops, foods, or humans; consume organic/nonorganic nutrients; and grow vegetatively, by expanding their habitats. In the presence of some stimuli, growing hyphae form thick-walled foot cells and branch to the aerial stalks. Swollen stalks successively form multinucleate vesicles, metulae, and phialides and finally develop conidiophores with immature conidia [15,18,19]. Then, the conidia on the conidiophores are matured and remain in the dormant stage. They transform their external structures, tolerate environmental stresses, maintain long-term viability, and prepare for the next stage of development through the activity of transcription and translation [20,21]. Under favorable conditions, quiescent conidia establish isotropic growth and germinate by producing the germ tubes. This lifespan of asexual spores may be delicately regulated by complicated and efficient mechanisms of genetic regulators.

This review describes the genetic regulatory factors involved in each developmental stage of *A. nidulans* conidia, from conidiogenesis, conidial maturation and dormancy to conidial germination. A comprehensive understanding of the functions of various regulators in the model organism *A. nidulans* can help prevent the formation of conidia that act as infectious particles, i.e., spores, or induce the maximal production of desired enzymes and proteins in filamentous fungi.

## 2. Research on *A. nidulans* Asexual Spores

Basic scientific studies on *A. nidulans* spores have been conducted for decades. To gain a broad understanding of various genetic regulators in conidia, we searched the keyword “*Aspergillus nidulans* spore” in the PubMed database. We obtained 648 articles published from 1969 to 2022, of which 200 studies explored the genetic regulatory factors in *A. nidulans* asexual spores. Studies on the regulators of *A. nidulans* conidia have been actively conducted worldwide, including in America, Asia, and Europe, with scientists primarily located in the USA, South Korea, and Germany (Figure 1).

*A. nidulans* conidia undergo a series of processes to form a specialized developmental structure known as the conidiophore. At the tips of conidiophores, the matured conidia remain in the dormant state and then start germinating and producing germ tubes through the activation of transcription and translation [18,21]. The life cycle of *A. nidulans* conidia is closely related to many regulators. To understand the development of *A. nidulans*, all of the regulatory factors, searched in PubMed (200 papers), were summarized by each developmental stage. Among them, we focused on the transcription factors, and how they act and organize their regulatory network. We also described major signaling pathways in *A. nidulans* conidia.

## 3. Conidiogenesis

Differentiated vegetative hyphae develop thick-walled foot cells and form conidiophores. An overview of the roles of regulators coordinating the formation of asexual structures and spores in *A. nidulans* is provided in Figure 2 and Table 1 and Table 2.

### 3.1. Temporal and Spatial Central Regulators of Conidiation

Approximately 50 years ago, the genes *brlA*, *abaA*, and *wetA* were morphologically investigated, which demonstrated that their appropriate temporal and spatial expression results in normal conidiophores formation in *A. nidulans* [36,37]. Scientists have named them central regulators of conidiation (Figure 2). BrlA is composed of 432 amino acids having two C_2_H_2_ zinc finger domains [38]. It has demonstrated that the *brlA* null mutant blocksin a conidial formation, exhibiting defective vesicles and loss of pigment [36]. The gene is named *brlA* (bristle-like) because the shape of the mutant resembles bristles. The *brlA* is primarily expressed in the early asexual development stage, and BrlA protein controls its own expression and that of various conidiation-related genes by binding to their promoters. The binding site is termed the BrlA-response element (BRE) and the sequence is 5′-(C/A)(G/A)AGGG (G/A)-3′ in *A. nidulans* [39].

One of the target genes directly regulated by BrlA is *abaA* (abascus). The *abaA* deletion mutant shows rod-like, aberrant conidiophores and fails to form proper metulae and phialides at intervals in place of chains of conidia [36,37,40]. As with *brlA*, it is named *abaA* because of the morphological characteristics of the mutant [41]. During the middle stage of conidiophore development, BrlA activates *abaA* mRNA expression. Then, AbaA, which has ATTS/TEA DNA binding domain factor and a potential leucine zipper, regulates its own expression and controls *brlAα* or other developmental genes (such as *wetA* and *vosA*) by recognizing the AbaA-response element (ARE) in their promoters [42]. The ARE is 5′-CATTCY-3′, where Y is a pyrimidine.

**Table 2 cells-12-01544-t002:** List of *Aspergillus nidulans* genes involved in the proper formation of asexual spores.

Name	Conidiogenesis	Description	Reference(s)
AclA/B	Activation	ATP-citrate lyase	[43]
AcoA~B	Activation	Aconidial genes, encoding putative aconitate hydratase	[44,45]
AflR	Activation	Sterigmatocystin/Aflatoxin Zn(II)_2_Cys_6_ transcriptional factor	[46]
AspE	Activation	*Aspergillus* septin E (septin protein)	[47]
BasA	Activation	Sphingolipid C4-hydroxylase, homolog of *S.cerevisiae* Sur2	[48]
Bud4	Activation	Bud site selection protein	[31]
CalB~H	Activation	Calcoflour hypersensitivity	[49]
CandA-C	Activation	Cullin-associated-nedd8-dissociated protein	[50]
ChsA	Activation	Chitin synthase encoding the fungal cell-wall integrity signaling (CWIS) pathway	[51]
FhbA/B	Activation	Flavohaemoglobins	[52,53]
GfsA	Activation	Galactofuranosyltransferase	[54]
GmcA	Activation	Glucose-methanol-choline oxidoreductase	[55]
KfsA	Activation	Kinase for septation	[56]
OdeA	Activation	Oleate Δ12 desaturases	[57]
PchA	Activation	Homolog of *S. cerevisiae* cyclin T	[58]
PclB	Activation	Homolog of *S. cerevisiae pcl* cyclins	[58]
PhnA	Repression	Phosducin-like protein (PhLP)	[59]
PhoA	Activation	PSTAIRE-containing kinase	[60]
PmtA/B	Activation	Protein O-mannosyltransferase involved in protein glycosylation	[61]
PmtC	Repression	Protein O-mannosyltransferase involved in protein glycosylation	[61,62]
PpoA/B	Repression	Psi-producing oxygenase involved in oxylipin biosynthesis	[63]
PpoC	Activation	Psi-producing oxygenase involved in oxylipin biosynthesis	[64]
PufA	Activation	Pumilio/fem-3 binding factor	[65]
SnaA~E	Activation	Suppressor of *nudA1*	[66]
StcE/J/U	Repression	Sterigmatocystin biosynthetic gene cluster	[46]
SdeA/B	Activation	Δ9-stearic acid desaturases	[67]
SumO	Activation	Small ubiquitin-like modifier involved in SUMOylation	[68,69]
UgeA	Activation	UDP-glucose-4-epimerase	[70]
VapA	Activation	FYVE-like zinc finger protein, one of the VipC-associated protein	[71]
VapB	Activation	H3-K9 specific histone methyltransferase, one of the VipC-associated proteins	[71]
VipC	Repression	H3-K9 specific histone methyltransferase, one of the Velvet interacting proteins	[71]
WscA/B	Activation	Homolog of *S. cerevisiae* Wsc1 involved in the fungal cell-wall integrity signaling pathway	[72]

In the late stage of conidiation, *wetA* (wet-white) is activated by AbaA. The null strain of *wetA* produces normal conidiophores, but it produces colorless, immature conidia in *A. nidulans* [36]. The *wetA* mutant conidia might undergo autolysis and show reduced viability. Moreover, the deletion strains of *wetA* have permeable conidia, whose wall layers are less condensed than those of the wild-type strain [73]. WetA has a conserved ESC1/WetA-related DNA binding domain that binds to the WetA-response element (WRE), 5′-CCGYTTGCGGC-3′ (Y = pyrimidine). The WetA, accumulated in conidia, recognizes WRE in the promoters of spore-specific genes and regulates their expression (*wA*, *yA*, *vosA*, and *atfB*) [74,75].

### 3.2. Upstream Developmental Activators (UDAs)

Previous studies have demonstrated the importance of UDAs in the development of *A. nidulans*. In 1994, the Adams group revealed that some developmental mutants formed cotton-like colonies with a “fluffy” morphology [76]. These included a mutant of *fluG* (fluffy locus A), and *flbA~E* (fluffy low *brlA* expression) genes. The *fluG* is required to produce an extracellular signal (a diorcinol-dehydroaustinol adduct) that initiates programmed asexual sporulation, and the *fluG*-derived signal can generate sparse conidiation and *brlA* expression [76]. Even the overexpression of *fluG* overcomes the developmental block and results in the formation of conidiophores in submerged culture [77]. In addition to *brlA*, the expressions of other early developmental regulatory genes (*flb* genes) are affected by FluG. The FluG-mediated developmental regulation is divided into two independent pathways; the activation of the Flb protein-mediated asexual development and the inhibition of FlbA-mediated vegetative growth (Figure 2).

#### 3.2.1. Flb Protein-Mediated Asexual Development

The transmitted FluG signal activates conidiation by derepressing SfgA-related pathways. SfgA, one of the suppressors of *fluG*, is a Gal4-type Zn(II)_2_Cys_6_ transcription factor. The *sfgA* null mutant exhibits hyperactive sporulation in liquid-submerged cultures, whereas overexpressed mutant exhibits the inhibition of conidiation. In other words, SfgA plays a role in the repression of asexual development [78]. Moreover, based on genetic analyses, SfgA has been demonstrated as a downstream factor of FluG but an upstream regulator of Flb proteins (*flbB*, *flbC*, *flbD,* and *flbE*, excluding *flbA*). As shown in Figure 2, FluG inhibits the expression of SfgA, counteracting the inhibitory effect of SfgA on Flb proteins. Coordinated by FluG and SfgA, four Flb proteins block hyphal growth and timely mediate the asexual development by positively affecting *brlA* expression [76]. Among these proteins, FlbB is a basic zipper-type transcription factor localized in the nucleus and apical extension (*Spitzenkörper*). The *flbB* null mutant exhibits blockage of the synthesis of an extracellular signaling compound, which is expressed in the early phases of vegetative growth for proper conidiation [79]. Further studies report that FlbB cooperates with other UDAs for regulating conidiation. First, FlbB physically interacts with FlbE and colocalizes at hyphal tips. FlbB-FlbE heterocomplex directly binds to the promoter of *brlA* to express *brlA* mRNA levels for proper asexual development. Furthermore, this complex can activate transcriptional levels of *flbD* for proper fungal development. FlbE is expressed throughout the life cycle of *A. nidulans* and has two conserved but hitherto uncharacterized domains [80]. The deletion and overexpression of *flbE* cause defective development, cell autolysis, and delayed *brlA* expression, indicating that the appropriate amount of *flbE* is crucial for proper fungal growth and asexual development [81]. Second, the FlbB complexes with FlbD, expressed by the FlbB-FlbE heterocomplex, which jointly bind to the *brlA* promoter for the activation of *brlA* expression [82]. FlbD is a c-Myb transcription factor, primarily found in the nucleus. Like other *flb* null strains, *flbD* absence mutant shows fluffy colonies and aconidial phenotypes, whereas the *flbD*-overexpressed strain causes inappropriate activation of *brlA* expression and the production of complex conidiophores [82,83]. Meanwhile, FlbC is not affected by other Flb proteins and acts independently for asexual development. FlbC has two C_2_H_2_ zinc fingers and one activation domain. FlbC is localized in the nucleus and binds to the promoters of *brlA*, *abaA,* and *vosA*, regulating their expression levels. Deletion of *flbC* causes delayed vegetative growth and reduced conidiation. Moreover, the overexpression of *flbC* results in restricted hyphal growth and reduced cellular activity. These indicate that modulated *flbC* is essential for balancing growth and development in *A. nidulans* [76,84]. 

#### 3.2.2. Inhibition of FlbA-Mediated Stimulation of Vegetative Growth

FlbA encodes 120 amino acids, having an RGS domain for the regulator of G protein signaling, which negatively regulates vegetative growth signaling [85]. Deletion of *flbA* results in unusual growth and abnormal conidiophores, whereas its overexpression causes hyphal tips to differentiate into spore-producing structures [76,86]. In other words, this protein plays an important role in mycelial proliferation and activation of asexual development. Commonly, RGS domain-containing proteins sense and respond to appropriate physiological and biochemical cues and function as GTPase-activating proteins in conjunction with α subunit of heterotrimeric G-proteins. As regulators of G proteins, they promote GTP hydrolysis of Gα to GDT, inactivating the G proteins (Figure 2). In *A. nidulans*, FadA is one of the Gα proteins, and the *fadA* deletion strain exhibits reduced growth without the impairment of sporulation [87]. In the heterotrimer status affected by FlbA, inactivated FadA (GDP-bound) exists with SfaD (suppressors of the *flbA*-loss of function mutations, Gβ) and GpgA (G protein gamma A, Gɤ) and represses vegetative growth [88]. When GTP is bound to FadA, this protein dissociates from the complex, and the separated FadA activates PkaA, which suppresses *brlA* levels, contributing to normal vegetative fungal growth and the inhibition of asexual development [89]. The remainders (GpgA and SfaD) separated from the complex also positively regulate hyphal proliferation and repress asexual developmental progression [90,91].

### 3.3. Upstream Regulators of BrlA

For hyphal cells to enter the stage of asexual development, appropriate temporal and spatial control of *brlA* expression must be applied in *A. nidulans*. Here, we describe the other upstream transcription factors of BrlA involved in the asexual development of *A. nidulans*, in addition to the UDAs that regulate *brlA* expression (Figure 2). StuA (stunted), studied in 1969, is known as a developmental modifier and is necessary for proper conidiophore formation [36]. StuA is one of the APSES transcription factors required for the transient and spatial regulation of *brlA* and *abaA* expression [92]. RgdA, named after retarded growth and development, is a putative APSES transcription factor that is important for the orderly organization of phialide formation [93]. RgdA affects *brlA* and *abaA* expression, but it is not influenced by *brlA* or *stuA*. RlmA, a *S. cerevisiae* ortholog of *rlm1*, is a MADS-box family transcription factor that is involved in proper *brlA* expression and phialide development, as well as in cell wall remodeling [94]. AslA, implicated in asexual differentiation with low-level conidiation, encodes a C_2_H_2_ zinc finger transcription factor that is related to normal asexual development and functions as an upstream activator of *brlA* [95]. MtfA is denominated from master transcription factor A, which encodes a C_2_H_2_ zinc finger domain. This protein has been demonstrated as a key factor for normal *brlA* expression, asexual development, and the production of several secondary metabolites in *A. nidulans* [96]. RcoA is a member of the WD repeat proteins that plays important roles in proper vegetative growth, asexual development, and carbon catabolite repression. This transcription factor also affects the expression of *brlA*, but not the signal transduction of *flbA* and *fluG* [97]. RtfA, encoding RNA-pol II transcription factor-like protein, regulates proper branched hyphal growth and conidiophore morphology by the accumulation of *brlA* transcript. Moreover, this protein affects the biosynthesis of several secondary metabolites, including sterigmatocystin and penicillin [98]. HbxA, a member of homeobox proteins, acts as an activator of asexual development and affects *brlA* mRNA expression. Deletion of *hbxA* results in aberrant asexual structures, whereas its overexpression results in enhanced production of conidiophores in liquid-submerged cultures [99]. 

Although Nsd (never in sexual development) proteins were previously described as transcription factors affecting the activation of asexual development, they are key negative regulators of conidiation. NsdC, with two C_2_H_2_ zinc fingers and a C_2_HC motif, is not only required for vegetative growth and sexual development but also negatively regulates asexual sporulation by repressing the *brlA* expression [100]. Like NsdC, one of the GATA transcription factors, NsdD, also affects conidiation by binding to the *brlA* promoter [101].

### 3.4. Other Key Regulators of Asexual Development

MedA (medusa) is also investigated in 1969 with StuA as mentioned above. This protein, which does not have any conserved domain, is essential for proper conidiophore formation [36]. MedA, known as an *Neurospora crassa* ortholog of *acon-3*, primarily modulates the expression of core conidiation genes (*brlA* and *abaA*) for the timely formation of metulae and phialides [102,103]. One of the WOPR fungi-specific DNA-binding proteins, OsaA (orchestrator of sexual and asexual development), indirectly controls conidiation by repressing downstream of the *velvet* regulator *veA*, which acts as a balancer between asexual and sexual development [104].

## 4. Conidial Maturation and Dormancy

Immature spores, formed at the ends of conidiophores, mature and stay in a dormant state. Dormant conidia modify the conidial wall, tolerate external stimuli, and maintain their viability for a long duration. The functions of controllers related to the maturation and quiescent phases in *A. nidulans* are shown in Figure 3 and Table 3.

### 4.1. The Velvet Family

Members of the *velvet* family are known as essential regulators of fungal growth, development, and secondary metabolism in ascomycetes and basidiomycetes [18]. They commonly share a “*velvet*” DNA-binding domain that is composed of 150 amino acids [119]. There are four *velvet* proteins, VeA, VelB, VelC, and VosA, in *A. nidulans*, which interact with each other or non-*velvet* regulators. Through the interaction of various combinations, *A. nidulans* can control development and conidiation in a temporal and spatial manner. Among the *velvet* proteins, VelB and VosA play pivotal roles in the maturation and dormancy of asexual spores (Figure 3). VelB (velvet-like protein B) acts as a positive regulator of asexual development and mediates spore viability, trehalose biosynthesis, and conidial pigmentation [120]. VosA (viability of spores A) also regulates tolerance to several stresses and the long-term viability of conidia and trehalose biogenesis [14]. Moreover, the VosA and VelB form a heterocomplex in conidia and play important roles in conidial maturation, cell wall composition, and spore germination. One example is that the VosA-VelB complex directly binds to the promoter of *fksA* and controls β-glucan biosynthesis in asexual spores [121]. This heterocomplex also modulates the expression of spore-specific structural and regulatory genes during conidiogenesis. Representatively, VadA (VosA/VelB-activated developmental gene A) is known as a gene affected by the VosA-VelB complex in conidia. VadA functions in conidial trehalose and β-glucan biogenesis, stress tolerance, spore viability, and germination in *A. nidulans* [122,123]. VadJ, regulated by the VosA-VelB complex, is one of the highly conserved sensor histidine kinases in *A. nidulans*. VadJ is required for the proper formation of asexual spores and maintenance of conidial viability [124]. Another Vad protein, VadZ, is a GAL4-like Zn(II)_2_Cys_6_ transcription factor that is essential for conidiogenesis and spore longevity [125].

In contrast, VidA (VosA/VelB-inhibited developmental gene A), repressed by VosA and VelB, has two C_2_H_2_ zinc finger domains at the C-terminus. This protein is involved in conidial trehalose and β-glucan biogenesis in *A. nidulans* [126]. There exists another Vid gene, *vidD*, which does not contain any known domains, but VidD is essential for normal fungal development, trehalose biosynthesis, and conidial long-term viability in *A. nidulans* [127].

### 4.2. Transcription Factors Involved in Conidial Maturation and Dormancy

McrA (Multicluster regulator A) is one of the putative GAL4-like Zn(II)_2_Cys_6_ transcription factors and is highly expressed in late asexual development. This protein affects proper conidiation by modulating *brlA* expression through the life cycle. Furthermore, McrA is known as a direct target of the VosA-VelB heterodimer and modulates long-term spore viability, trehalose and β-glucan biogenesis, and proper conidial pigmentation [128]. One of the VosA-controlled regulatory genes, *sclB* (sclerotia-like), contains a Zn(II)_2_Cys_6_ zinc cluster fungal-type DNA binding domain. Unlike VosA, which represses the premature induction of *brlA*, SclB induces the early activators of asexual development (FlbC, FlbD, and BrlA) and influences conidiogenesis. In addition, SclB plays important roles in conidial viability and tolerance to oxidative stress [129]. Another Zn cluster family member, *zcfA* encodes a Zn(II)_2_Cys_6_ zinc finger protein and is known as a putative VosA target gene in conidia. The deletion of *zcfA* results in increased asexual spore formation and induced mRNA levels of *brlA*. Phenotypic analysis of Δ*zcfA* conidia shows that ZcfA plays a key role in conidial viability, trehalose biogenesis, and thermal stress resistance [130]. HbxB, one of the homeobox family members, is highly expressed in asexual spores and modulates the production of asexual spores and conidial stress resistance to thermal, oxidative, and UV stresses [99]. As a transcription factor, HbxB affects the transcriptomic levels of various genes, regulating trehalose biosynthesis, and β-glucan degradation in conidia [131]. CsgA is a GAL4-like Zn(II)_2_Cys_6_ transcription factor specifically expressed in *A. nidulans* conidia. The deletion of *csgA* results in an increased number of conidia, and *csgA*-deleted conidia exhibit augmented trehalose contents and increased tolerance to thermal, oxidative, and UV stresses compared to WT conidia. In addition, CsgA is required for normal conidial viability and germination [132].

### 4.3. Genetic Regulators Related to Stress Tolerance in Conidia

#### 4.3.1. Mitogen-Activated Protein Kinase (*MAPK*) Cascades

During maturation, the cellular composition of immature spores is altered by several regulators, and consequently, they possess the ability to withstand various external stresses. Representatively, the histidine-to-aspartate (His-Asp) phosphorelay systems actively function so that spores can survive for a long time even under extreme environmental conditions. In *A. nidulans*, NikA is a histidine-specific protein kinase (HK) that first recognizes stimuli. NikA plays important roles in proper conidial reproduction and conidial resistance to certain fungicides and osmotic stress. In the presence of stimuli, NikA responds and activates downstream stress-related regulators [133]. Upon activation by NikA, YpdA transmits the signals to the response regulator, SskA. The deletion of *sskA* results in defective asexual development, conidial viability, and sensitivity against cold and oxidative stresses (H_2_O_2_). SskA phosphorylates SskB (MAPKKK), which subsequently phosphorylates PbsB (MAPKK) and the ortholog of Hog1 (MAPK). There are two homologs of *S. cerevisiae* Hog1 in *A. nidulans* and other *Aspergilli*: SakA (stress-activated MAP kinase) and MpkC. Both of these homologs physically interact with the upstream regulator PbsB, but they have the same or different functions in *A. nidulans* [134]. As a member of the Hog1/Sty1/p38 family, SakA is a well-known key regulator of spore stress tolerance in filamentous fungi. Similarly, in *A. nidulans*, the deletion mutant of *sakA* exhibits defective conidial production and Δ*sakA* is sensitive to thermal, oxidative, and cell wall stresses. Conversely, the deletion of *mpkC* results in increased conidial production and resistance to oxidative stress. However, both Hog1 homologs, SakA and MpkC, are essential for the long-term survival of conidia [135]. Another MAPK, MpkB, is phosphorylated by other MAPK components (pheromone MAPK pathway), and activated MpkB is also involved in fungal development. MpkB, regulated by VosA, is pivotal for spore viability and inhibits conidial germination. MpkB also influences fungal autolysis [136]. The other MAPK, MpkC, is activated by the fungal cell wall integrity signal (CWIS). Phosphorylated MpkC modulates proper conidial germination as well as conidial cell wall integrity by upregulating genes related to α-, β-1,3-glucans and chitin biosynthesis [137].

#### 4.3.2. Other Transcription Factors Related to Stress Tolerance in Conidia

SrrA is one of the response regulators and components of a stress-sensing phosphorelay system in *A. nidulans*. Like SskA, SrrA is activated by YpdA. However, unlike SskA, SrrA directly translocates into the nucleus. As a specific transcription factor, SrrA affects conidial resistance against oxidative stress by regulating antioxidant genes such as *catB*. Furthermore, SrrA plays important roles in conidial formation and spore viability [138]. AtfA, an ortholog of *Schizosaccharomyces pombe atf1*, is a member of the activating transcription factor/cAMP-responsive element-binding protein (ATF/CREB) family. AtfA, containing a bZIP domain, permanently exists in the nucleus and responds to oxidative stress during spore development. When SakA accumulates in the conidial nucleus by oxidative stress, it physically interacts with AtfA, regulating the expression of antioxidant-related genes in conidia. Consequently, AtfA plays pivotal roles in the conidial antioxidant response (tBOOH and H_2_O_2_) and long-term viability [139,140]. Recent research shows that AtfA interacts with another bZIP transcription factor, AtfB, for coordinating asexual development and stress tolerance [140]. Although AtfA appears to be more important than AtfB in the oxidative stress defense system of *A. nidulans* spores, AtfB is also essential for tolerating thermal and oxidative stresses. Another transcription factor related to oxidative response is NapA, which is an *S. cerevisiae* Yap1 functional homolog and contains a bZIP domain. Similar to the functions of AtfA, NapA is involved in oxidant detoxification (menadione and H_2_O_2_). When stimulated by oxidants, NapA protects by positively regulating both nonenzymatic (e.g., glutathione and thioredoxin) and enzymatic (e.g., catalases and superoxide dismutases) pathways in conidia. Moreover, NapA regulates asexual development and carbon utilization [141]. RsrA (regulator of stress response) is known as a C_2_H_2_ zinc finger transcription factor that is required for fungal growth and sporulation. RsrA directly represses antioxidant genes such as *glrA*, *trxA*, and *catB* as well as NapA in the presence of reactive oxygen species such as tBOOH and H_2_O_2_ [142].

## 5. Conidial Germination

Under appropriate conditions, the resting conidia break the quiescent state and modify their morphology. Through the alteration of cell wall composition and molecular organization, they swell and produce the germ tube (polarized growth). The germinated spores can expand their habitats such as humans, animals, and plants. Therefore, it is important to understand the regulatory mechanisms related to conidial germination. Although there are several coordinators mediating conidial germination (Figure 4 and Table 4), we focus on the genetic regulators and transcription factors.

### 5.1. Transcription Factors Involved in Germination

CreA is a Cys_2_His_2_ transcription factor for *carbon catabolite repression* (CCR). In the presence of glucose as a carbon source, CreA directly or indirectly represses genes encoding enzymes (cellulases, xylanases) for degrading alternative carbon sources. However, in the absence of glucose, CreA is ubiquitinated and targeted by the proteasomes. As a result of the degradation of CreA, enzymes for alternative carbon sources are biosynthesized in *A. nidulans* [179]. In addition to CCR, the Δ*creA* strain shows defective spore germination [180]. SocA, a Zn(II)_2_Cys_6_ transcription factor, was first discovered by mutagenesis and isolation of FLIP (Fluffy in Phosphate) mutant. The null mutant of *socA* shows deficient colony growth and abnormal morphogenesis as well as an altered germination pattern [62].

### 5.2. Other Regulators Related to Germination

The cAMP-PKA pathway (PKA pathway) is closely related to *A. nidulans* conidial germination. When GanB, which is Gα forming a heterotrimer with SfaD (Gβ) and GpgA (Gγ) mentioned earlier, separates from the complex and then activates CyaA, the activated CyaA as the adenylate cyclase produces cAMP from ATP, which attaches to PkaR and promotes conidial germination [169]. In addition, PkaB, a secondary protein kinase A catalytic subunit that is separated from PkaR by cAMP binding, promotes conidial germination and spore resistance to oxidative stress and inhibits conidiation while remaining alone [181]. Meanwhile, the dissociated GanB and SfaD-GpgA dimers are reunited by RgsA (regulator of G-protein signaling family), which indirectly affects asexual development and inhibits vegetative growth. RgsA plays a vital role in proper conidial germination and tolerance to thermal and oxidative stresses [182].

## 6. Conclusions

The genus of *Aspergillus* is one of the most abundant filamentous fungi in the air. They proliferate by producing a number of asexual spores, which constitute the major reproductive mode. Floating at short and long distances, conidia take root in the appropriate environment or host and grow by changing their morphology. They undergo asexual development and produce conidiophores. The matured conidia on the conidiophores stay in the resting phase and prepare for the next generation. These processes are coordinated by several genetic regulators or signaling pathways, which have been investigated by researchers. In this review, we summarized the key genetic regulators and their roles in each stage of asexual reproduction in the model organism *A. nidulans*. This information will help us gain a better understanding of the organizational and systematic developmental process and may help prevent the development of pathogenic spores or maximize the production of desired ones. Nevertheless, scientific studies must be continuously and deeply scrutinized as there are still several unexplored regulators.

## Figures and Tables

**Figure 1 cells-12-01544-f001:**
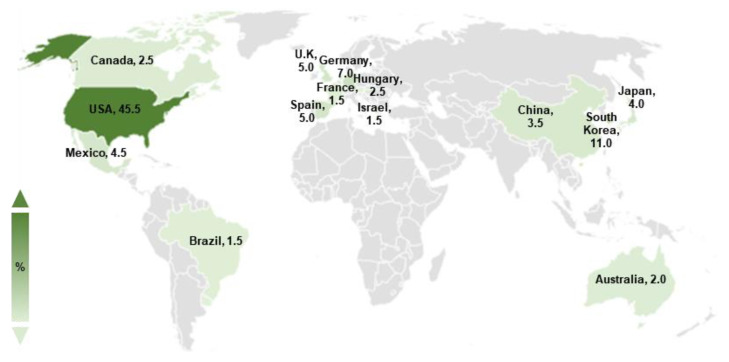
Studies on *Aspergillus nidulans* asexual spores. The worldwide *distribution* of studies on *A. nidulans* spores. The darkness of color indicates a high ratio of published articles. Only countries where the ratio of studies on *A. nidulans* spores is >1% are indicated.

**Figure 2 cells-12-01544-f002:**
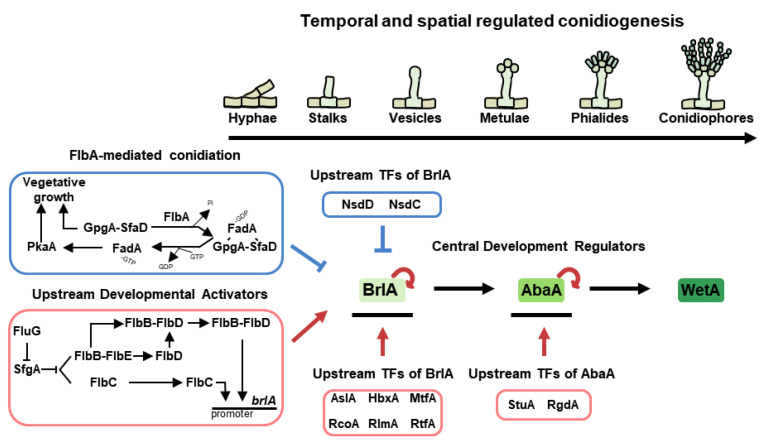
Asexual development in *A. nidulans*. The temporal and spatial central regulators of conidiation in *A. nidulans* lifespan. Each regulator affects the expression of BrlA as upstream developmental activators (UDA) and upstream regulators of *brlA*.

**Figure 3 cells-12-01544-f003:**
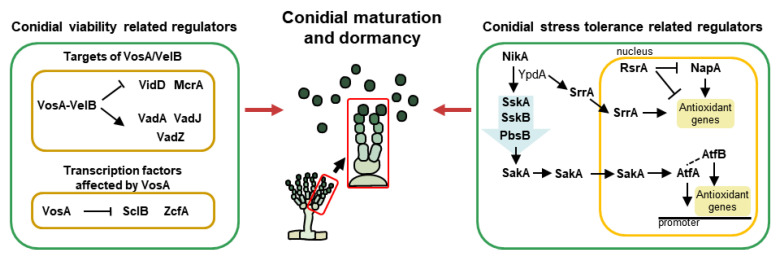
The genetic regulators of maturation and dormancy in *A. nidulans* conidia. A simplified model for conidial maturation and dormancy, including *velvet* proteins, transcription factors, and MAPK-mediated regulators.

**Figure 4 cells-12-01544-f004:**
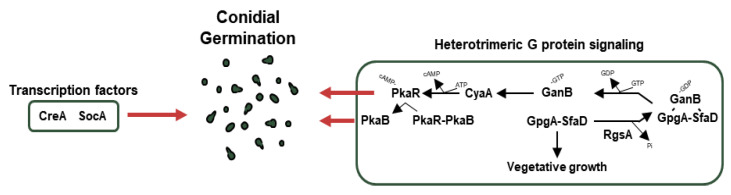
Genetic regulators of germination in *A. nidulans* conidia. A schematic presentation of the genetic regulators of conidial germination.

**Table 1 cells-12-01544-t001:** List of *Aspergillus nidulans* genes involved in different asexual developmental stages.

Name	Stages	Description	Reference(s)
DnfD	Vesicles	Phospholipid flippase, homolog of *S. cerevisiae* Drs2-Neo1-Family Neo1	[22]
PabA	Vesicles	Putative regulatory subunit of protein phosphatase 2A (PP2A)	[23]
DopA	Vesicles, Metulae	Dopey protein	[22]
NimX	Vesicles, Metulae	Cyclin-dependent kinase involved in cell cycle control	[24]
GmtA/B	Metulae	Putative GDP-mannose transporter	[25]
HymA	Metulae	Hypha-like metulae	[26]
BemA	Metulae, Phialides	Protein kinase activator	[27]
FigA	Metulae, Phialides	Putative low-affinity calcium family protein	[28]
GcnE	Metulae, Phialides	SAGA complex histone H3K9 acetyltransferase catalytic subunit	[29]
ParA	Metulae, Phialides	Putative regulatory subunit of protein phosphatase 2A (PP2A)	[23]
UgmA	Metulae, Phialides	UDP-galactopyranose mutase	[30]
Axl2	Phialides	Axial budding positional marker	[31]
PclA	Phialides	Homolog of *S. cerevisiae pcl* cyclins	[32]
SnaD	Phialides	Coiled-coil protein associated with the spindle pole body, named after the suppressor of *nudA1*	[33]
SthA/B	Phialides	Sthenyo	[34]
TcsA	Phialides	Two-component signaling protein involved in oxidative/osmotic stress pathway	[35]

**Table 3 cells-12-01544-t003:** List of *Aspergillus nidulans* genes involved in conidial maturation and dormancy.

Name	Conidiogenesis	Conidial Viability	Conidial Stress Response	Description	Reference(s)
CatA			Resistant to H_2_O_2_	Spore-specific catalase A, involved in oxidative/osmotic stress	[105]
CatB			Resistant to H_2_O_2_	Catalase B, involved in oxidative/osmotic stress	[106]
CchA	Activation		Sensitive to CFW and CR	Calcium channel	[107]
CpsA	Activation		Resistant to MSB, SDS and CFW	Capsule polysaccharide synthase	[108]
DewA				Spore-wall fungal hydrophobin, named after detergent wettable	[109,110]
DlpA			Resistant to heat and H_2_O_2_	Dehydrin-like protein	[111]
DnjA	Activation	Maintenance	Resistant to heat	Putative DnaJ proteinRegulation of trehalose biosynthesis	[112]
HmbB	Activation	Maintenance		Putative high-mobility group box proteinRegulation of trehalose biosynthesis and proper germination	[113]
LysB/D			Resistant to heat, UV and H_2_O_2_	Homoisocitrate dehydrogenase/synthase	[114]
MidA	Activation		Sensitive to CFW and CR	Stretch-activated calcium channels, named after mating-induced death	[107]
MonA	Activation	Maintenance	Resistant to heat	A subunit of a guanine nucleotide exchange factor Regulation of trehalose biosynthesis	[115]
MpdA			Resistant to heat Sensitive to benomyl	Mannitol-1-phosphate 5-dehydrogenase	[116]
MtlA	Activation		Resistant to CFW and CR	Mid2-like protein	[117]
PufE	Repression	Maintenance	Resistant to heat	Pumilio/fem-3 binding factor Regulation of trehalose biosynthesis	[65]
RodA				Rodlet protein composed of fungal spore wall	[110]
TpsA		Maintenance	Resistant to heat and H_2_O_2_	Trehalose-6-phosphate synthase Regulation of trehalose biosynthesis	[118]

**Table 4 cells-12-01544-t004:** List of *Aspergillus nidulans* genes involved in asexual spore germination.

Name	Conidiogenesis	Germination	Description	Reference(s)
ArgB	Activation	Activation	Ornithine carbamoyl transferase, homolog of *S. cerevisiae* Arg3 Conidial resistant to heat, UV, and H_2_O_2_	[114]
ApsA	Activation	Activation	Anucleate primary sterigmata protein A	[143,144]
ApsB	Activation	Activation	Anucleate primary sterigmata protein B	[144]
AspA	Activation	Activation	*Aspergillus* septin A	[145]
AspB	Activation	Activation	*Aspergillus* septin B	[146]
AspC	Activation	Activation	*Aspergillus* septin C	[145]
CalA	Activation	Activation	Fungal thaumatin-like proteins, named after calcoflour hypersensitivity	[49,147]
CaM		Activation	Calmodulin involved in calcium-calcineurin signaling	[148]
CetA		Activation	Fungal thaumatin-like proteins, named after conidial-enriched transcripts	[147]
ChiA		Activation	Chitinase involved in fungal cell-wall integrity signaling (CWIS)	[149]
CmkA/B		Activation	Ca^2+^/caM-dependent protein kinase involved in calcium-calcineurin signaling	[150]
ConF/J		Activation	Conidiation-specific gene Conidial sensitive to polyol	[151]
CotA	Activation	Activation	NDR protein kinase, homolog of *S. cerevisiae* Cbk1	[152,153]
CpcB	Activation	Activation	Gβ-like protein, named after Cross pathway control WD repeat protein B	[154]
DnfA	Activation	Activation	Phospholipid flippases, homolog of *S. cerevisiae* Drs2-Neo1-Family Dnf1/2	[22,155]
DnfB	Activation	Activation	Phospholipid flippases	[22,155]
FphA		Activation	Fungal phytochrome (red light-sensing photoreceptor)	[156]
GapA	Activation	Activation	Ras GTPase-activating protein	[157]
GcsA		Activation	Glucosylceramide synthase	[158]
GlrA		Activation	Glutathione reductase	[158]
GprH		Activation	G protein-coupled receptor	[159]
HmbA	Activation	Activation	High-mobility-group B protein A	[160]
LkhA	Activation	Activation	LAMMER kinase, homolog of *S. pombe* Lkh1	[161]
MobB	Activation	Activation	Homolog of *S. cerevisiae* Mob2	[153]
NimA		Activation	Cell-cycle regulated serine/threonine protein kinase, homolog of *S. pombe* Kin3	[162]
NpgA	Activation	Activation	4′-phosphopantetheinyl transferase, named after the null pigmentation mutant	[163]
PexC PexE~G	Activation	Activation	Peroxisome biogenesis protein (peroxin)	[164]
PclA		Activation	Phosphatidylinositol phospholipase, homolog of *S. cerevisiae* Pcl1 Conidial resistant to cold	[165]
PrsB/C		Activation	Phosphoribosyl pyrophosphate synthetase	[166]
RasA		Activation	Ras-like protein	[167]
RicA	Activation	Activation	GDP/GTP exchange factor in Heterotrimeric G protein signaling	[168]
SchA		Activation	Sch9-like kinase	[169]
SepA		Activation	Formin protein involved in the formation of an actin ring at the septation site	[170]
SodVIC		Activation	α-COP-like protein, named after the stabilization of disomy	[171]
SvfA	Activation	Activation	Homolog of *S. cerevisiae* survival factor 1 Conidial resistant to cold, MSB and H_2_O_2_	[172]
PmtA (=SwoA)		Activation	Protein O-mannosyltransferase (Swollen cells) Conidial resistant to CFW	[173,174]
SwoB	Activation	Activation	Swollen cells Conidial resistant to CFW	[174]
TeaC		Activation	Cell end marker protein	[175]
UgtA	Activation	Activation	UDP-Galf transporter	[176]
WspA	Activation	Activation	Wiskott–Aldrich syndrome protein	[177]
YpkA	Activation	Activation	Polyphosphate kinase, homolog of *S. cerevisiae* Ypk1	[178]

## Data Availability

Not applicable.

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
