# Peer review of "Regulators of the Asexual Life Cycle of Aspergillus nidulans"

_cells, 2023, doi:10.3390/cells12111544_

Round 1

Reviewer 1 Report

This review presents a comprehensive description of regulators the asexual life cycle of Aspergillus nidulans. The authors summarized  the genetic regulatory factors involved in each developmental  stage of A. nidulans conidia, from conidiogenesis, conidial maturation and dormancy to
conidial germination.   This manuscript provides a comprehensive understanding of the functions of various regulators in the model organism A. nidulans can help prevent the formation of conidia that act as infectious particles, i.e., spores, or induce the maximal production of desired enzymes
and proteins in filamentous fungi.

Overall, this manuscript is well written and should be  published.

Minor points:

1 p1, line 18, foot sell should be foot cell

2 Figure 3, vosA controlled gene sclB, VosA should be an inhibitor of SclB from the published paper (Thieme et al., PloS Genetics, 2018)

Author Response

Overall, this manuscript is well written and should be published.

⇒ We really appreciate this positive comments. Based on this reviewer’s comments, we revised our manuscript. We hope that this revised version can be published in Cells.

Minor points:

1. p1, line 18, foot sell should be foot cell

⇒ Thank you for this valuable comment. Following this comment, we changed it

Lines 18-19: Each A. nidulans conidiophore is composed of foot cell, stalk, vesicle, metulae, phialides, and 12,000 conidia.

2. Figure 3, vosA controlled gene sclB, VosA should be an inhibitor of SclB from the published paper (Thieme et al., PloS Genetics, 2018)

⇒ Thank you for this valuable comment. Following this comment, we revised Figure 3.

Reviewer 2 Report

Regulatory mechanism of conidiation in filamentous fungi is quite complex. This review paper summarises it very well in an easily accessible manner. This review paper will be very helpful to the reader's understanding of the conidiation regulation and will contribute to the advancement of research in this field. One very minor point is that it would be better if "Zn2Cys6" is written as "Zn(II)2Cys6".

Author Response

We really appreciate this positive comments. Based on this reviewer’s comments, we revised our manuscript. We hope that this revised version can be published in Cells.

⇒ Following this comment, we replace “Zn2Cys6" with "Zn(II)2Cys6” through this manuscript.